# Mechanical interactions among followers determine the emergence of leaders in migrating epithelial cell collectives

Medhavi Vishwakarma[1,2], Jacopo Di Russo [1,2], Dimitri Probst[3], Ulrich S. Schwarz[3], Tamal Das[1,2,4] & Joachim P. Spatz [1,2]

Regulating the emergence of leaders is a central aspect of collective cell migration, but the underlying mechanisms remain ambiguous. Here we show that the selective emergence of leader cells at the epithelial wound-margin depends on the dynamics of the follower cells and is spatially limited by the length-scale of collective force transduction. Owing to the dynamic heterogeneity of the monolayer, cells behind the prospective leaders manifest locally increased traction and monolayer stresses much before these leaders display any phenotypic traits. Followers, in turn, pull on the future leaders to elect them to their fate. Once formed, the territory of a leader can extend only to the length up-to which forces are correlated, which is similar to the length up-to which leader cells can transmit forces. These findings provide mechanobiological insight into the hierarchy in cell collectives during epithelial wound healing.

[1] Department of Cellular Biophysics, Max Planck Institute for Medical Research, Jahnstraße 29, 69120 Heidelberg, Germany. [2] Department of Biophysical Chemistry, Heidelberg University, INF 253, 69120 Heidelberg, Germany. [3] Institute for Theoretical Physics and BioQuant, Heidelberg University, INF 267, 69120 Heidelberg, Germany. [4] TIFR Centre for Interdisciplinary Sciences (TCIS), Tata Institute of Fundamental Research Hyderabad, 500107 Hyderabad, India. Correspondence and requests for materials should be addressed to T.D. (email: tdas@tifrh.res.in) or to J.P.S. (email: spatz@mr.mpg.de)

Collective cell migration drives many critical biological processes including wound healing, organogenesis, and cancer development[1–6]. Effective collective migration, in many cases, requires formation of leader cells at the tissue boundary[3,7,8]. As an illustration, in the well-studied collective migration of a monolayer of epithelial cells mimicking wound healing, leader cells display a large lamellipodial structure at the front edge, move ahead of other non-leader or follower cells, and provide the guidance for the migrating group[4,7–10]. Beyond wound healing, leader cell formation has also critical influences on branching morphogenesis[11] and metastatic invasion[12] both in vivo and in vitro. The dynamics that dictate leader-cell selection is a central problem[4,6,7], yet remains largely elusive, even for epithelial monolayers that show clear leader-follower organizations. It is not understood, for example, at what level, interface or bulk monolayer, the signals for formation of leader cells originate, and why only a fraction of cells at the front becomes leader cells. Many recent studies revealed diverse molecular and biomechanical traits of the leader cells[8,13,14] and many discovered interesting biophysical aspects of epithelial mechanics[6,15–19]. While most of these studies describing formation and regulation of leader cells essentially attribute them to the interfacial properties[17,18,20], some also indicate the possibility that the dynamics of leader cells could eventually be regulated by the dynamics of the bulk[4,17,18]. However, the nature of the contribution from bulk in the selection of leader cells remains mostly unknown as the events occurring at the onset of or preceding the leader cell formation remain obscure. Recent evidences showing long-lived traction patterns extending to several cell diameters[21] and velocity patterns[14], even in a confluent monolayer[22], indicate the relevance of group dynamics in epithelial monolayers, but if and how the group dynamics might control the selection and regulation of leader cells at the margin remains unclear.

To understand how leader cells emerge, here we have studied mechano-biological aspects of epithelial wound healing resolved in time and space. Using traction force and monolayer stress microscopy, we find that the leader cells at the wound-margin are effectively selected by the mechanical interactions of the follower cells located behind the leading edge. We demonstrate that the followers mechanically pull on the future leader, aiding in their polarization and protrusion. Combining experimental data with theoretical modeling, we are able to show that the territory of a leader extends only to the length up to which forces are correlated in the monolayer, which is similar to the length up to which leader cells can transmit forces. This finding, therefore, provides a mechanism for the formation of leader cells during collective cell migration, wherein we place mechanical interactions between the cells as the central player that determines when and where a leader cell would emerge.

## Results

**Emergence of leader cells.** To study the time evolution of the wound margin in a controlled and quantitative manner, we grew confluent monolayers of Madin–Darby Canine Kidney (MDCK II) epithelial cells within confined areas and then lifted off the confinement to prompt two-dimensional sheet migration (Fig. 1a). This in vitro model mimics the process of re-epithelialization in wounded skin tissues[23]. Initially for ~30–45 min, cells at the wound-margin did not exhibit any noticeable lamellipodial protrusion, which we name Phase 0 (Fig. 1b, c). After this phase, leader cells with prominent lamellipodial protrusions started emerging at the wound margin (Phase 1, Fig. 1b, c). As the migration progressed, each of these leader cells generated an outgrowth in the wake (Fig. 1c, Supplementary movie 1). This phase 1 lasted for about three hours. After this

time, additional leaders started emerging from the existing outgrowths leading a new group of followers (Phase 2, Fig. 1b, c). We also observed this biphasic behavior in an uncontrolled wound assay by scratching the cells in a confluent monolayer (Supplementary Fig. 2, Supplementary movies 2, 3). We then wondered whether emergence of leader cells (between Phase 0 and 1 and between Phase 1 and 2) is an autonomous decision of each leader cell or requires preparatory structural rearrangement at the margin or within the bulk monolayer. To this end, we hypothesized that owing to the mechanobiological integrity of cytoskeletal elements at both single and multicellular levels, any structural change would lead to detectable variations in cell–matrix traction stresses. Hence, elucidating the cell-matrix traction stress landscape with traction force microscopy would enable us to identify any zone going through structural reorganization. To this end, we grew the monolayer of MDCK cells over a soft polyacrylamide substrate containing fluorescent beads. Subsequently, much before the cells showed any sign of noticeable migratory activity (Supplementary fig. 1, Supplementary movie 7), we observed local heterogeneities in the cell–matrix traction stress that correlated with the emergence of leader cells (Fig. 1c, d). These regions of high forces were seen to be more prevalent within 2–6 cell layers behind the prospective leader cells than any other randomly chosen location of similar layer depth parallel to wound margin (Fig. 1d, Supplementary movies 4–6). Interestingly, such changes in traction also appeared before the emergence of second set of leaders, at the transition point from phase 1 to phase 2, for a particular outgrowth (Fig. 1c, d, Supplementary movies 4, 5). This observation strongly indicated a general correlation between the traction force activity in follower cells and the emergence of leader cells. As revealed by monolayer stress microscopy, these zones behind the prospective leader cells also developed elevated tensile stresses, at both temporal phases, just before the emergence of leader cells in front of them (Fig. 1c–e, Supplementary movie 6). We also independently observed similar preparatory elevation of traction and monolayer stresses among the follower cells for HaCaT cells, which are immortalized human keratinocytes (Supplementary Fig. 3a–c), pointing out the general relation between follower and leader cells for leader cell identification.

Next, to understand the possible implications of the local increase in traction and tensile stresses on leader cell formation, we examined different microscopic mechanobiological traits of the individual cells including cell shape and aspect ratio, before the commencement of migration process. Relevantly, even a fully confluent epithelium can exist in two distinct structural states: jammed (solid-like) and unjammed (fluid-like) as explained by dynamic heterogeneity of the monolayer[24–26]. Moreover, as elucidated in normal and asthmatic bronchial epithelia, an unjammed monolayer displays considerably higher cell-matrix traction than a jammed one[25]. Further, a non-dimensional shape index ($q$) measuring the ratio of the perimeter to square root of the projected cross-sectional area ($q = P/\sqrt{A}$) of the individual cells can capture the specific state of the cells. Jammed cells were characterized with $q < 3.81$ and unjammed cells with $q > 3.81$[24,25]. In order to probe if the increase in traction stresses are associated with the possibility of local unjamming within a pre-migratory monolayer, we computed the shape index ($q$) for each individual cell. Indeed, follower cells behind the prospective leader cells displayed higher shape index than other cells within the same depth, at both temporal phases (Fig. 1c–f). Taken together, these results indicated a systematic elevation of tensile stresses and unjamming-like transitions of the cells immediately behind the prospective leader already in pre-migratory monolayer, though confirmation of true unjamming transition would ideally require further characterization of force fluctuations and perimeter elasticity. Since in an epithelial monolayer, the tensile stress is

transmitted across the cell–cell junctions and is exclusively balanced by the cell–cell adhesive stress[24], these results also implicated a local increase in the cell–cell junctional stresses between future leaders and corresponding follower cells.

We then investigated whether, beyond the aforementioned correlation, this local increase in traction and tensile stresses and the unjamming-like behavior among the follower cells has any causal relationship with the cells ahead of them becoming the leader cells. Our earlier research has elucidated that a tumor-suppressor protein, merlin, supports collective cell migration by regulating the polarization of a migration promoting molecule Rac1 and thereby governing the direction of the cell motility

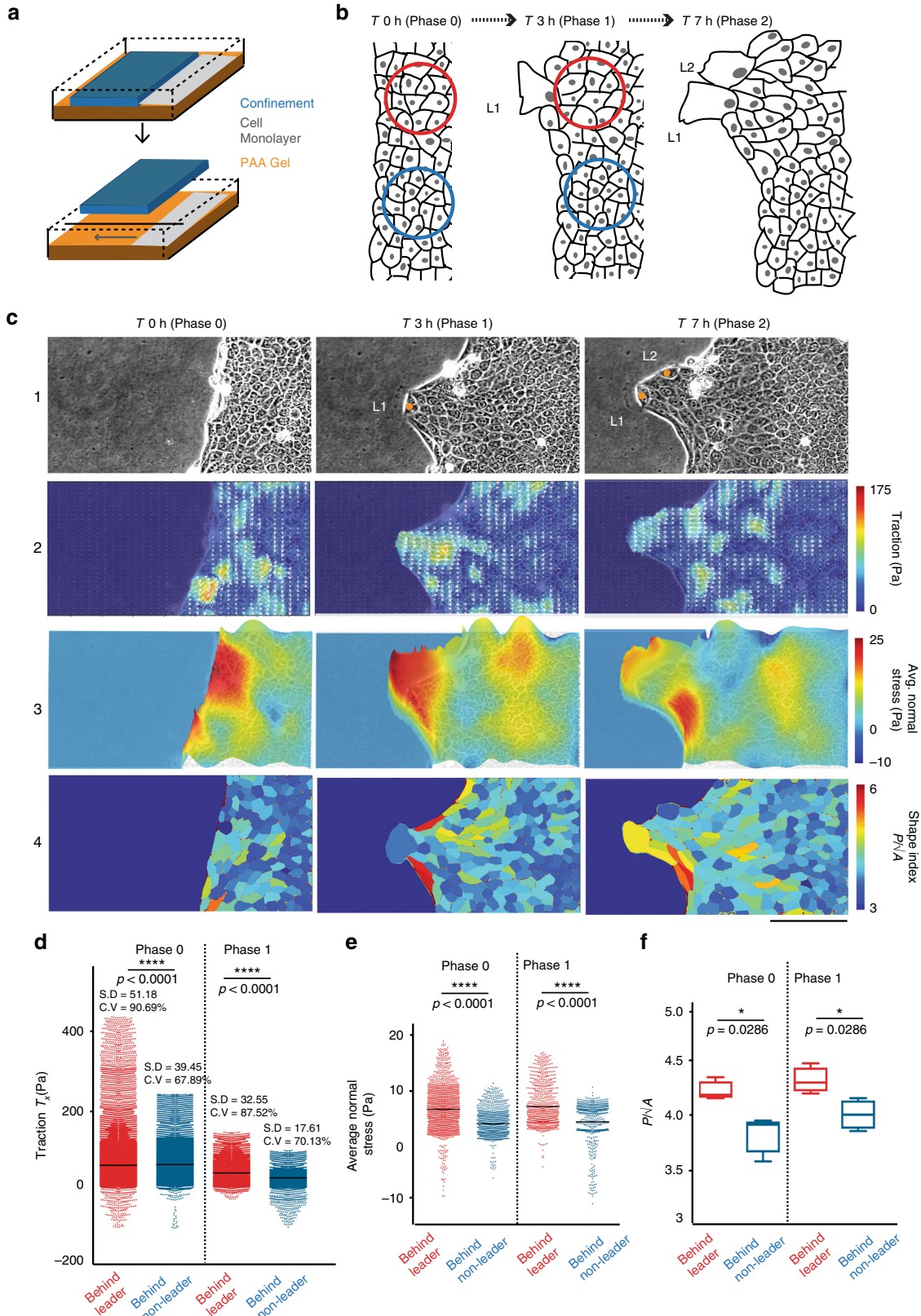

within a monolayer[23]. Incidentally, on depleting the expression level of merlin with a specific small interfering RNA (siRNA), the resultant cell phenotype appeared intrinsically more unjammed and less correlated with their neighbors than the control cells as characterized by particle image velocimetry (Supplementary Fig. 4). The inverse relation between unjamming and velocity correlation length was similar to previous study[27]. In contrast to wild-type cells, merlin-depleted cells stay perpetually in an unjammed-like state even in confluent condition (Supplementary Figs. 4, 5a). Merlin-depleted cells showed higher traction stresses as well as higher shape index than control cells (Supplementary Fig. 5a). Taking advantage of this property of merlin depletion, we then mixed fluorescently labeled merlin-depleted cells with unlabeled wild-type cells in 1:10 ratio and determined the probability of the emergence of a leader cell in front of any merlin-depleted cell groups (Supplementary Fig. 5b). The statistics took account of only those cases where the 2–6 layers behind the wound margin contained at least two merlin-depleted cells. We further excluded any merlin-depleted leader cell. For control experiments, a scrambled siRNA replaced merlin siRNA while other conditions remained unchanged. The results consequently revealed that the presence of relatively unjammed merlin-depleted cells increases the probability of a marginal cell ahead of them to become a leader cell (Supplementary Fig. 5c).

These experiments together revealed that local development of contractile stresses as shown by increased stresses in the monolayer and unjamming-like transition in the following layer as shown by the increased shape indices stimulate leader cell formation during collective migration of epithelial cells. They also implied that while the emergence of leader cells is itself an interfacial phenomenon[18], the factor regulating it have a hitherto unknown non-interfacial or bulk component, originating from the dynamics of the cells located behind.

**Distance between leader cells.** While introducing merlin-depleted cells introduces controlled variation in the cell dynamics, in a genetically homogeneous monolayer, peaks in stress landscape appear spontaneously in a stochastic manner. In fact, as depicted by monolayer stress microscopy and also described previously[28], the stress distribution within the bulk epithelial monolayer manifests a rugged landscape with peaks and basins extending over several cell diameters (Fig. 2a). The landscape also evolves with time and thus reveals dynamic heterogeneity within the bulk monolayer, which is reminiscent of the spatially heterogeneous dynamics in dense colloidal suspension approaching glass transition[26,29]. Interestingly, the cellular shape index, $q$, also shows similar spontaneously emerging heterogeneity, and its value appears to be distributed on both sides of the transition point ($q = 3.81$) even in an apparently quiescent and packed epithelial monolayer[25]. Since the emergence of the leader cells was preceded by the appearance of high stress regions behind them (Fig. 1c), we presumed that the extent to which the stress

propagates across the monolayer also described by the stress correlation length, $F_{CL}$ (Fig. 2a), should closely match the separation between adjacent leader cells at the wound margin (Fig. 2b). Conceptually, the characteristic length-scale of the spatial autocorrelation function, C(r), of the average normal stress, known as the force correlation length[30] (Supplementary Fig. 10), indicates the average number of cells that could collectively integrate their forces through the cell-cell junctions[28] and give rise to the observed ruggedness in the stress landscape (Fig. 2a). Indeed, the distribution of leader-to-leader ($d_{LL}$) separation distance showed excellent correspondence with the distribution of force correlation length ($F_{CL}$) ($d_{LL} = 162.4 \pm 30.2$ μm; $F_{CL} = 170.5 \pm 38.7$ μm; mean ± s.d.) in both MDCK and HaCaT cells (Fig. 2c). This result further validated that in a genetically homogeneous monolayer, the stochasticity in mechanical activity of the bulk monolayer indeed determines the apparently random emergence of leader cells at the interface.

With these results in view, we next examined to what extent the contribution from the mechanobiology of bulk monolayer can prevail when the monolayer is presented with a perturbation at the interface. To this end, we generated monolayers with highly curved beak-shaped regions, using a soft-lithography based patterning technique (Fig. 2e, Supplementary Fig. 6). These high curvature beaks lead to the generation of locally confined high tractions at the margin and thus, impose an interfacial bias in the force landscape towards leader cell generation[20]. By varying the spacing between two consecutive beaks, we controlled the length-scale of the interfacial bias. In spite of the imposition of interfacial bias, the final distribution of leader cell separation in patterned monolayers appeared very similar to that in non-patterned monolayer ($d_{LL}$ for non-patterned (unbiased): $162.4 \pm 30.2$ μm; 75 μm pattern: $143.3 \pm 22.3$ μm; 300 μm pattern: $166.2 \pm 31.8$ μm; mean ± s.d.; Fig. 2d, Supplementary movie 8). Together these results established the importance of the collective cellular dynamics in regulating the emergence of the leader cells at the interface and indicated that bulk-mechanobiological parameters such as the length-scale of force transduction could control the length-scale of leader cell emergence.

**Modifying the force correlation length.** Subsequently, to test our hypothesis that force transduction determine the distance between the leader cells, we modulated the length-scale of force transduction by both chemical and physical methods. For chemical modification, we used the widely used pharmacological means of controlling the actomyosin contractility by treating the cells with a non-muscle myosin-II inhibitor, blebbistatin (5 μM), or a myosin-light-chain phosphatase inhibitor, calyculin A (1 nM). Blebbistatin reduced the contractile forces, while calyculin increased it. Then, as expected, blebbistatin treatment enhanced the ruggedness of the stress landscape and lowered the force correlation while calyculin treatment regularized the stress landscape and increased the correlation length (Fig. 3a, d), both

**Fig. 1** Force transmission from followers facilitate leader cell formation. **a** Illustration depicting generation of confinement on a polyacrylamide (PAA) gel using polydimethylsiloxane (PDMS) blocks. Removal of confinement triggers collective cell migration. **b** Schematic representation of cell monolayer immediately after (T 0 h), three hours (T 3 h), and seven hours (T 7 h) after confinement removal. **c** From top to bottom, representative phase contrast images (Panel-1), corresponding traction force profiles (Panel-2), landscapes of average normal stress overlapped with phase contrast images (Panel-3) and corresponding color coded maps of shape indices (Panel-4) showing accumulation of high forces and local unjamming-like transition in followers behind future leaders in different phases of wound healing. **d** Scatter dot plot showing traction forces behind leader and non-leader cells in phase 0 and phase 1. **e** Scatter dot plot showing average normal stress behind leader and non-leader cells in phase 0 and phase 1. **f** Box plot showing shape indices in the regions behind leader and non-leader cells in phase 0 and phase 1. Box plots shows median and quartiles. L1 and L2 are leader cells appearing in Phase-1 and Phase-2, and are marked with orange dots. Red and blue circles mark the zones considered for statistical comparison of the traction stress behind leaders and non-leaders respectively and are defined by force localization length (radius = $L_P$/2). Whiskers are maximum and minimum data points. Line in scatter dot plots represent mean. *$P < 0.05$, ****$P < 0.0001$, Mann–Whitney test. S.D is standard deviation and C.V. is coefficient of variation. Scale bars = 100 μm. Data collected from at least three independent experiments

in comparison to the control case. Remarkably, in each case, the force correlation length matched the corresponding leader-to-leader distance (Fig. 3a–c, Supplementary Fig. 8, Supplementary movie 9). HaCaT cells also showed similar trends of changes in leader-to-leader distance (Supplementary Fig. 7). Complementing the chemical perturbation, we also altered the force correlation

length by physical means, by changing the stiffness of substrate over which the cells migrated (Fig. 3c, right panel). In this case, both force correlation length and leader-to-leader distance increased with increasing stiffness of the substrate (Fig. 3a–d). Together, these results confirmed systematic mechano-biological regulation of leader cell generation during collective migration of

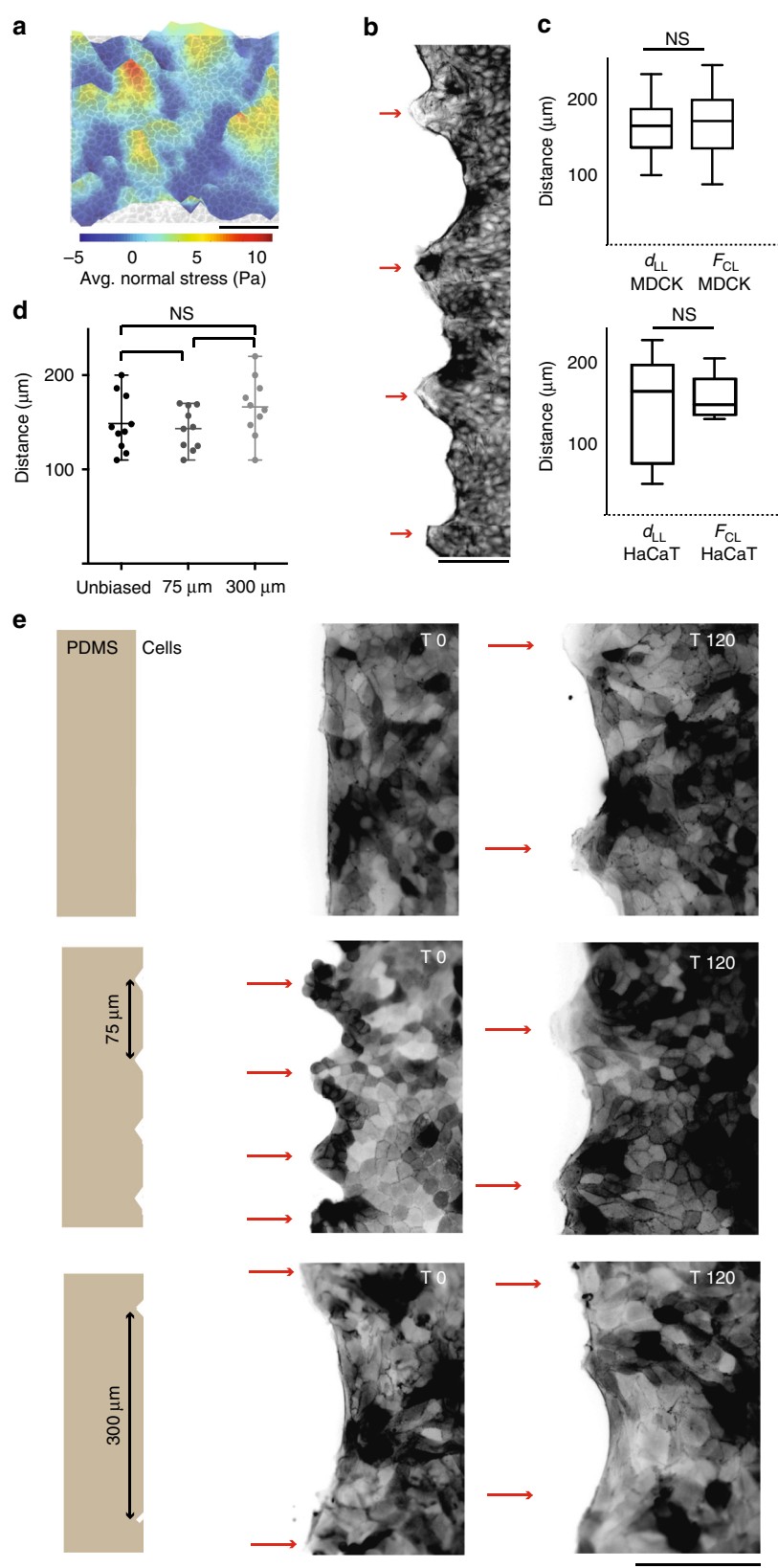

epithelial cells in wound closure, where a system-property emerging out from the bulk such as the force correlation length dictates the emergence of leader cells.

**Territory of a leader cell in time and space**. We then asked whether the length scale of force transduction could quantitatively define the transition from Phase 1, where an outgrowth is dominated by one leader to Phase 2, where multiple leaders per outgrowth could be seen (Supplementary movie 1). Velocity flow data show that in migrating monolayer, followers that ends up in the fingers behind the leader extend up-to 200 μm from the tip[22], demonstrating the role of leader cell as a local guide that also exert high traction force[8]. Therefore, we presumed that the transition from phase-1 to phase-2 would happen when the number of followers per leader exceed the extent up-to which the leader cell can transmit forces. Subsequently, we computed the followers per leader by calculating the Euclidean distance between the velocities of followers, $\mathbf{v}$ and velocity of the leader cell, $\mathbf{v_L}$ (Fig. 4a). Cells for which the Euclidean distance $|\mathbf{v}-\mathbf{v_L}|$ is lower than the threshold $|\mathbf{v_L}|$ are considered to have a velocity magnitude and direction similar to the leader cell and are thus classified as followers. Spatial extent of these followers can then be estimated from the profile of Euclidean distance (Fig. 4a). To compute the length up-to which leader cell transmit force, we used a theoretical approach, where we modeled the epithelial layer, as a thin elastic plate of height $h_c$, elasticity $E_c$, Poisson's ratio $v$ and with an isotropic contraction stress $\sigma_0$ reflecting cellular contractility[31–33]. This layer is further coupled to the underlying substrate elastically via springs of stiffness density $Y$ (Fig. 4b). Then, we obtained a localization length, $L_{P,}$ incorporating contributions from both bulk and substrate parameters[32]. It can be defined as the length up to which a localized force originating from the leader cell is transmitted (see methods). By using default values as measured previously[34] for cell and substrate parameters (Supplementary table 1), we arrived at a localization length $L_P \approx 168$ μm, which reflects a spatial limitation up to which a localized force is transmitted. Interestingly, the spatial extent of followers per leader was not only in agreement with the distance up-to which forces are correlated in the monolayer (Fig. 4c), but was also in agreement with the distance up to which a leader cell can transmit force in the bulk ($L_P$) as calculated theoretically (Fig. 4c, green line). Further, a time plot displayed that a plateau is reached when followers per leader exceed this distance (Fig. 4d, green curve), thereby marking a transition to phase-2 where exceeding number of followers is balanced by formation of new leaders. Intuitively, modification in force correlation length by Blebbistatin or Calyculin-A, as also verified theoretically by obtaining a plot between localization length ($L_P$) and cell elastic modulus ($E_C$), showing $L_P$ as a function of $E_C$. (Supplementary Fig. 9) subsequently modified the number of followers per leader and the transition time between these phases (Fig. 4d, red curve, blue curve). These results demonstrate that the length up to which force is transmitted from the leader to follower during migration dictates the onset of Phase 2.

In order to explain why the distance between leader cells in phase-1 and the spatial extent of followers in phase-2 is limited to $L_P$ we used our mechanical model to simulate the experimental situation. We simulated leader cell formation by applying two-point forces separated by the distance $d_{LL}$, we then varied $d_{LL}$ and measured the horizontal distance $|X_C – X_B|$, where $X_C$ is the horizontal displacement of the boundary between leader cells while $X_B$ is the baseline displacement (Fig. 4e). Here, the two leader cells are distinguishable when $X_C – X_B \cong 0$. We found out that relative horizontal displacement $|X_C – X_B|$ is an exponentially decaying function of the distance with the characteristic decay length equivalent to the force localization length $L_P$ (Fig. 4f). This indicates, that in Phase-1, two leader cells are distinguishable when separated by at least distance $L_P$. For Phase 2, we modeled a leader cell at the tip of a cellular cohort of radius $R$ and measured the effect of a point force $\mathbf{F_L}$ exerted by the leader on the remaining bulk, i.e., the relative displacement of the bulk point $U_C$ with respect to the leader cell, $U_L$ (Fig. 4e). Interestingly, relative displacement of $U_C$ also appeared as exponential functions of the distance with the characteristic decay length equivalent to the force localization length $L_P$ (Fig. 4f). This indicates that when the cellular cohort grow larger than $L_p$, such as in Phase-2, additional follower cells are not registered to belong to the cohort anymore and therefore a new leader is identified to accommodate the increasing number of followers. These trends of relative horizontal displacement $|X_C– X_B|$ in phase-1, and relative displacement of $U_C$ in phase-2, was also seen upon tuning cellular stiffness by drugs (dotted lines). Together, theoretical modeling and simulation results provide reasoning for the experimentally observed distance between leaders, $d_{LL}$ in Phase 1 and for the need of an outgrowth to transit to Phase 2 by placing the force localization length, $L_P$ as a central player.

**Followers pull on the future leader**. Finally, we aimed to directly demonstrate that pulling forces from followers stimulate leader cell formation. As demonstrated previously for single cell migration, pulling on the rear end of a cell cause the cell to protrude forward[35], we attempted to extend and demonstrate this hypothesis in the collective cell migration perspective. To this end, we designed a cell-pulling experiment wherein we cultured cells under confined conditions on an elastic silicone substrate, containing a micrometer wide trench. As described previously[23], upon uniaxial stretching of this substrate, the cells residing on one side of the trench would exert pulling force on their neighbors on the other side (Fig. 5a). In this case, the trench was made about 100 μm away from the cell monolayer (Fig. 5b). The confinement was lifted off to allow migration until a single row of cells cross the trench at various locations, after which, cell pulling was performed by stretching with an impulse strain (25% s$^{-1}$), as described previously[23] (Fig. 5b). Importantly, we observed that upon stretching, marginal cells which experienced the pulling force from behind, started forming lamellipodia like protrusions towards the front (Fig. 5c, middle panel, Supplementary movie 10). However, these protrusions were retracted again after

**Fig. 2** Length scale of force transmission override interfacial bias and determine territory of a leader cell. **a** Representative landscape of average normal stress in bulk (here shown with 3D height) overlaid on corresponding phase contrast image showing groups of cells under each peak. **b** Actin staining image showing appearance of leader cells (shown by red arrows) at an interval $d_{LL}$ in phase 1. **c** Box plots showing no significant difference between leader to leader distance ($d_{LL}$) and the Force correlation length ($F_{CL}$) in both MDCK (top) and HaCaT (bottom) cells. **d** Statistical distribution of $d_{LL}$ two hours after the confinement removal (T = + 90 min) for different micro-patterned and non-patterned (unbiased) monolayers. **e** design of PDMS micro-stencil, demonstrating beak shape interfacial patterns to bias leader cell formation (left). Representative images of migrating LifeAct-transfected MDCK cells right after (T = 0 h, middle) and two hours after (T = 2 h, Right) the confinement removal, for different micropatterns. Whiskers in box plots are maximum and minimum data points. Scatter dot plots display mean with maximum and minimum range as error bars. NS: not significant, $P > 0.05$, Student's $t$-test. Scale bars, 100 μm. Data collected from at least three independent experiments

~60 min, likely due to the pulling force from the already existing leaders (Fig. 5c, right panel, Supplementary movie 10). This forced-induction of leader cells, therefore, strongly demonstrated that the collective and integrated pulling force from the follower cells has the ability to stimulate leader cell formation immediately in front of the former, further consolidating our hypothesis

related to the unique role of follower cells in the selection of leaders. Importantly, since this forced-induction of leader cells also resulted in transient breaking of the peripheral actin cables (Fig. 5c, middle panel, Supplementary movie 10), it also places the force exerted from the follower to leader cells at the upstream of actin cable-breakage, while enumerating the possible mechanisms

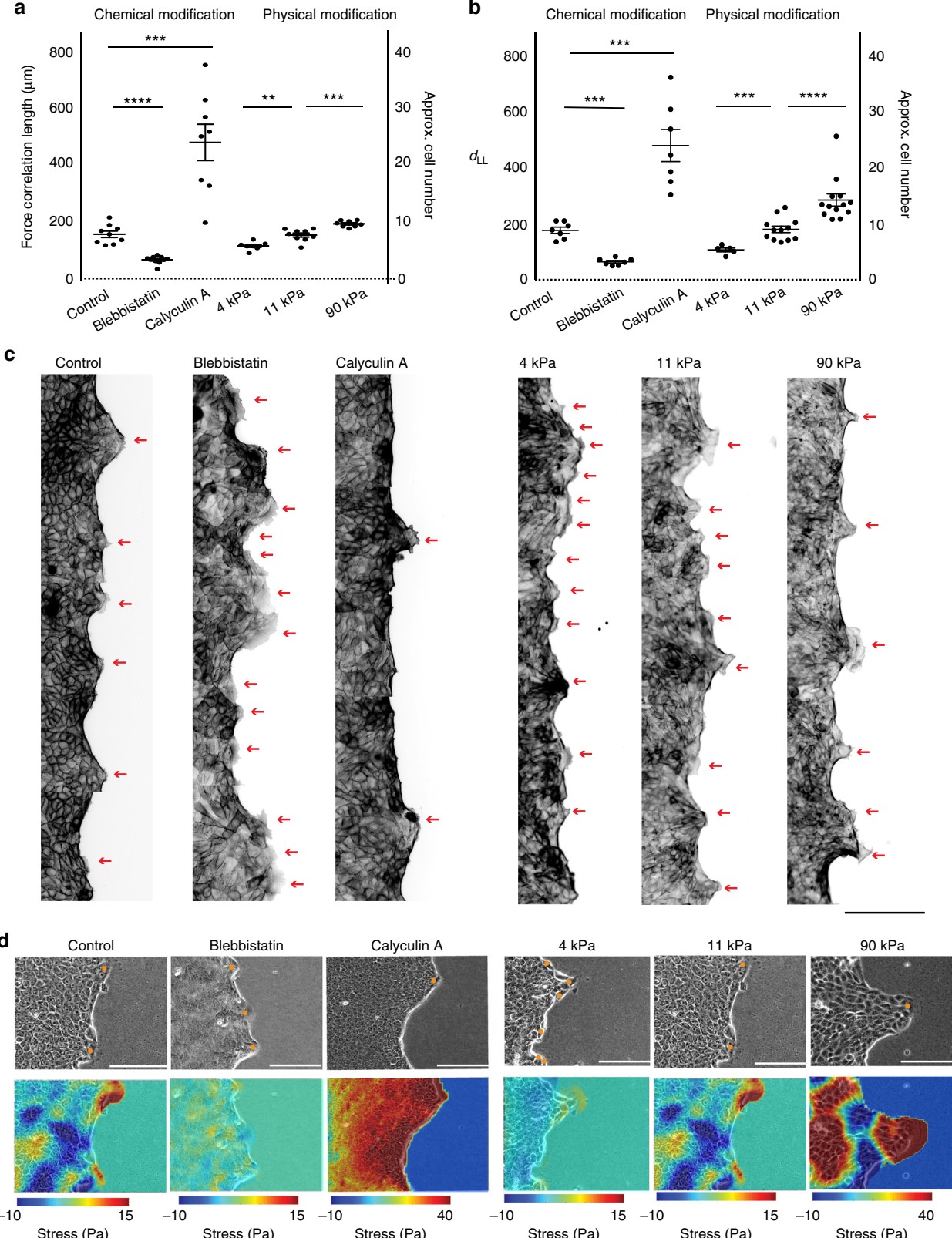

behind the leader cell formation. Finally, the fact that this "induced" leader retained the lamellipodial structure only transiently, showed the importance of the length-scale of the cellular force transduction, indicated by the bulk force correlation length, which eventually controlled the spatial dynamics of leader cells.

## Discussion

Taken together, we propose a non-cell autonomous mechanism underlying the emergence of leader cells at epithelial wound margin. Owing to the dynamic heterogeneity and force imbalance in the monolayer, followers pull on the future leader at the wound margin, thereby facilitating its polarization and protrusion formation (Phase 0). Once the leader cell is formed, its role as a puller plays an important role (Phase 1) and therefore, formation of another leader is prevented in a territory that is defined by how far the force from the leader is transmitted in the monolayer (as defined by $L_P$), beyond which new leaders are formed (Phase 2). Owing to its non-interfacial and mechano-biological nature, the mechanism of leader cell formation that we describe here is fundamentally distinct from the canonical perspective of interfacial regulation of leader selection. This does not exclude the possibility that feedback between interfacial curvature and motility will further stabilize leader cell identity[17,18], as also supported by recent studies on the curvature sensing ability of epithelial cells[36]. Our results, however, support a view in which such effects are downstream of the mechanical selection of leader cells by their future followers (Figs. 2e, 5, Supplementary movies 6, 8, 10, 11). Furthermore, upstream or downstream to the biophysical route that we provide here, there might exist molecular routes which are involved in cellular polarization and leader cell formation, that are still to be discovered. The work presented here, therefore, not only demonstrates the importance of mechanical interactions in the interplay between leader and follower cells, but also indicates that the molecular, mechanical, and interfacial components greatly influence each other and are not mutually exclusive.

## Methods

**Cell culture**. Madin-Darby canine kidney cells (MDCKII, Health Protection Agency) were cultured in minimal essential medium (MEM, Sigma) supplemented with 2 mM L-glutamine (Invitrogen), 10 U ml$^{-1}$ penicillin, 10 µg ml$^{-1}$ streptomycin (Pen Strep, Invitrogen), and 5% fetal bovine serum (FBS, Invitrogen). Human keratinocytes line (HaCaT, Cell Lines Service) were cultured in high glucose dulbecco's modified eagle medium (DMEM, Gibco) supplemented with GlutaMax$^{TM}$, 10% FBS, 10 U ml$^{-1}$ penicillin and 10 µg ml$^{-1}$ streptomycin.

**Micro-patterning**. Polydimethylsiloxane (PDMS) stencil masks with holes of defined shapes were fabricated in an adapted soft lithography process[20,37]. Briefly, desired shapes of monolayers were designed in a QCAD program and transferred on transparencies using a high-resolution printer (JD Phototools). In a clean room facility, SU-8/25 negative photoresist (MicroChem, Newton, MA, USA) was spin-coated on a 2″ silicon wafer to a final thickness of about 50 µm. The wafer was then baked on a hot plate for 5 min at 65 °C followed by a second baking for 15 min at 95 °C. The transparencies containing the "photographic negative" of the pattern to be transferred were used as masks to illuminate the wafer with UV light for 12 s in Mask Aligner MBJ4 (Suess MicroTec Lithography, Munich, Germany). To remove

the unexposed photoresist, wafers were immersed in SU-8 Developer mr-Dev600 (Microresist Technology, Berlin, Germany). The prepared wafers containing the desired geometric pattern were then treated with 1 H,1 H,2 H,2H-Perfluorooctyl-trichlorosilane to reduce surface adhesiveness. A sandwich consisting of the patterned wafer, 0.5 ml of uncured PDMS, a piece of parafilm, a piece of paper and a glass slide was put into a custom made molding press to obtain uniform pressure distribution. PDMS was pressed against the wafer in order to create thin PDMS membrane containing holes of desired shapes. The assembly was put into a compartment dryer at 65 °C for 100 min to allow PDMS polymerization. To prevent cell adhesion, prepared stencil masks were incubated in a solution of Pluronic F-127 (Sigma Aldrich, 2% w/v in deionized water) for 30 min prior to use.

**Migration experiments and Traction force microscopy**. For performing collective cell migration in defined patterns, PDMS microstencils with patterned holes or ibidi cell culture inserts (80209) were allowed to stick onto the customized glass bottom dishes (5 cm diameter) coated with 10 µg ml$^{-1}$ fibronectin unless otherwise specified. Cells were seeded into the dish and incubated in a cell-culture incubator for 1 h until they adhere onto the fibronectin-coated glass accessible through the holes of the microstencils. The unattached cells were then removed by replacing the media. The set up was incubated again overnight or until a confluent cell monolayer of around 3000 cells/mm$^2$ is obtained after which, the PDMS stencil was removed to trigger collective migration. Migration experiments were carried out at either 37 °C and in 5% CO$_2$ environment either inside a stand-alone cell culture incubator, or within an incubator staged over the microscope.

Traction force microscopy was performed as described previously[38]. Briefly, glutaraldehyde activated glass bottom dishes (MatTek) were used to cast thin polyacrylamide (PAA) gel substrates (Young's modulus of about 11 kPa) containing 0.5 µm fluorescent carboxylated polystyrene beads. These gel surfaces were then functionalized with sulphosuccinimidyl-6-(4′-azido-2′-nitrophenylamino) hexanoate (Sulfo-SANPAH, Thermo Scientific) and covalently coated with 0.5 mg ml$^{-1}$ fibronectin (Sigma) to ensure cell attachment. A horizontal confinement was created on the functionalized PAA gels using thin PDMS blocks. Cells were seeded in the confined areas and grown until a confluent monolayer is obtained. Subsequently, confinement was released by lifting off the PDMS block and images for cells and beads were acquired as the cells migrated. After experiment, cells were trypsinized and resulting bead positions in relaxed state were obtained (i.e., reference images). The displaced images were aligned to correct for drift and compared to the reference image using particle image velocimetry to create a regular field of displacement vectors with a grid spacing of 5.44 µm. Displacement vectors were interpolated using cubic splines. From these vectors, traction stresses were reconstructed using regularized Fourier Transform Traction Cytometry[39] with a regularization parameter chosen by Generalized Cross Validation[40].

**Monolayer stress microscopy**. Stresses within the monolayer were then calculated from the cell-substrate tractions using a force balance algorithm written in MATLAB (MathWorks) as described in our previous study[23]. Force correlation length was computed by characteristic length scale of the spatial autocorrelation function of the average normal stresses as formulated elsewhere. Briefly, the extent to which the force propagates across the monolayer was obtained by the characteristic length-scale of the spatial autocorrelation function, $C(r)$, of the average normal stress, which is known as the force correlation length[30]. $C(r)$, was calculated as:

$$C(r) = \frac{1}{N var(\acute{\sigma})} \sum_{i,j=1}^{N} \sum_{|r_i - r_j| = r} \delta \acute{\sigma}_i \cdot \delta \acute{\sigma}_j$$

where $\delta \acute{\sigma}_i$ is the local deviation of the average normal stress at position $r_i$ from its spatial mean $\sigma'_i$ and $var(\sigma')$ is the variance of these deviations. Stress correlation length was determined as the point where the stress correlation function became negligible in value. For practical purposes, we took correlation length as the distance at which the correlation function was equal to 0.01

---

**Fig. 3** Modifying length scale of force transmission modifies distance between leaders. **a** Scatter dot plot showing force correlation length in terms of cell number upon chemical modification, i.e., by treating the cells with blebbistatin and calyculin A compared to the control, and upon physical modification, i.e., by changing substrate stiffness. **b** Scattered dot plot showing distance between leaders in terms of cell number upon chemical and physical modifications. **c** Representative images of control, blebbistatin treated, calyculin A treated collectives and collectives on gels with varying stiffness 4 KPa, 11 KPa, and 90 KPa (from left to right) as stained for actin, show changing distance between leader cells upon chemical modification and physical modifications. **d** Representative phase contrast images (top panel) and landscapes of average normal stress from control blebbistatin treated, calyculin A treated collectives and from collectives on gels with varying stiffness, 4 KPa, 11 KPa and 90 KPa (from left to right), as measured during migration. Note that the Calyculin A treated and 90KPa stress profiles have a different scale due to high stress levels. Cell density was kept constant = 3000 cells/mm$^2$. Line represents mean and errors bars represents S.E.M. **** $P < 0.0001$, Mann–Whitney test. Scale bars, 100 µm. Data collected from at least three independent experiments

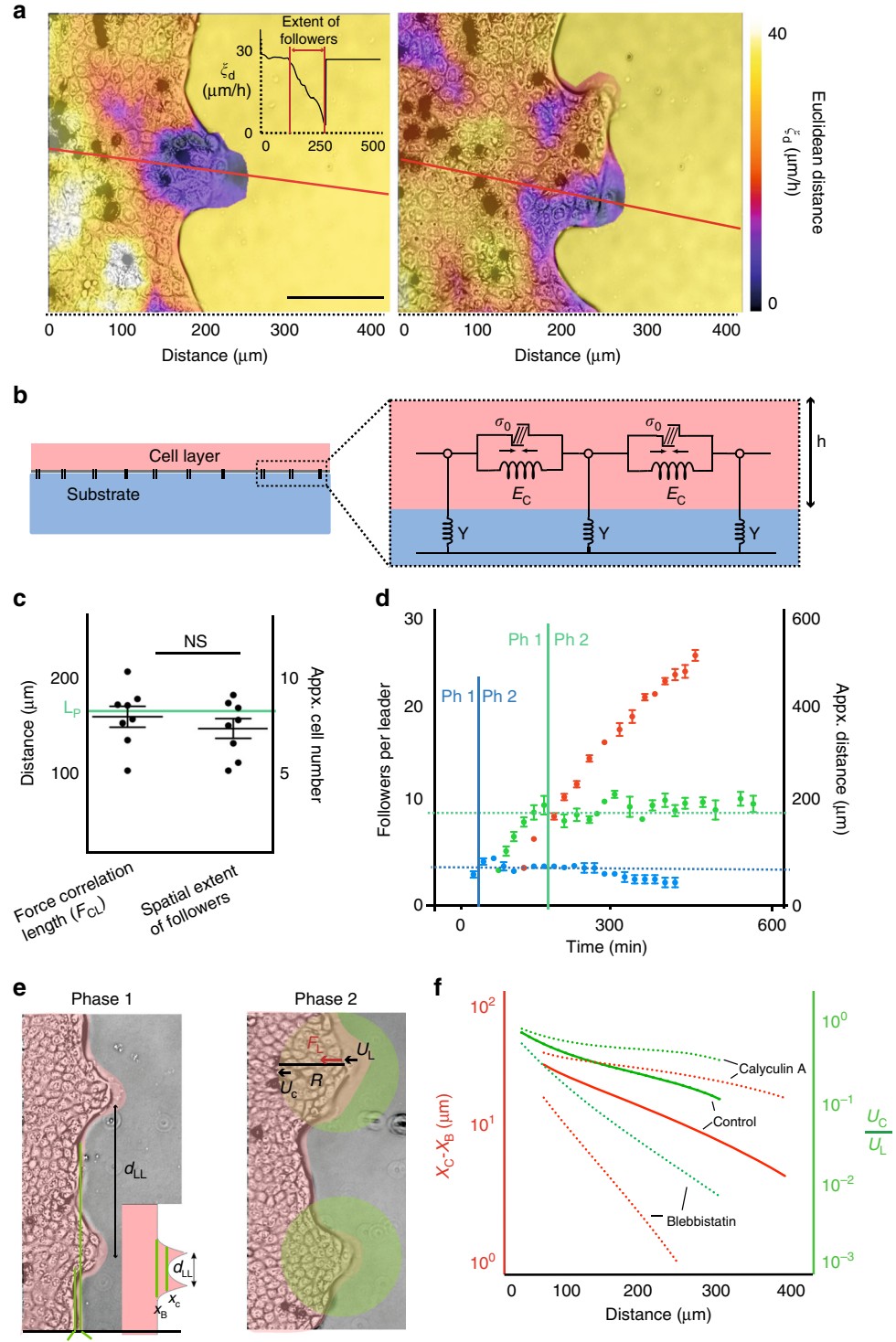

**siRNAs and transfection**. Merlin siRNA was purchased from Qiagen (5′ CAAAGAGAGGGAGACAGCCTTGGAT -3′) and was transfected by reverse transfection using Lipofectamine 2000 (Invitrogen), as instructed by the manufacturer. The scrambled siRNA was purchased from Qiagen.

**Inhibition studies and Immunostainings**. Blebbistatin, an inhibitor of myosin II, and calyculin A, a phosphatase inhibitor, were obtained from Sigma. Powders of these drugs were dissolved in DMSO (Sigma) to make the stock. Before removing the confinement, cells were treated with 5 μM blebbistatin and 1 nM calyculin A respectively in Opti-MEM reduced serum medium for 1 h at 37 °C in a 5% $CO_2$ humidified incubator. During migration, Opti-MEM was replaced by MEM containing 5% FBS, 2 mM L-glutamine and the respective inhibitor. For actin stainings,

cells were fixed and permeabilized before adding Alexa fluor-488 labeled phalloidin (Thermo Fisher Scientific) for visualization of the actin cytoskeleton.

**Cell pulling experiment**. A customized cell-pulling device was used to apply mechanical strain to an elastomeric PDMS substrate, onto which cells were later cultured, as also described previously[23].The PDMS chamber for cell culture, was made by casting PDMS in a Plexiglas mold at an elastomer to crosslinker ratio of 10:1. The PDMS was cured for 2 h at 65 °C. The chamber was then peeled, sonicated in 70% ethanol and oxidized in an oxygen plasma environment for 1 min. This process coated a thin solid film of silica on top of the flexible PDMS membrane. A narrow trench was generated on the surface by scratching the silica-coated membrane with a micro-needle (tip size 20 μm) under an inverted table top

**Fig. 4** Length up-to which force is transmitted regulates number of followers and necessitate the transition from phase-1 to phase-2. **a** Spatial extent of followers per leader as computed by Euclidean distance calculated along the red line (Inner plot). Phase contrast images overlaid with Euclidean distance profile showing extent of followers. **b** side view of the model used to calculate length scale of force transmission ($L_P$). Epithelial layer of height $h$, modeled as an elastic medium of elasticity $E_C$, Poisson's ratio $\nu$ and an isotropic contraction stress $\sigma_O$. **c** Scatter dot plot showing spatial extent of followers in sync with force correlation length, $F_{CL}$ as calculated experimentally and force localization $L_P$ as calculated theoretically. Line represents mean and errors bars represents S.E.M. **d** Transition from phase-1 (Ph1) to phase-2 (Ph2) in a developing outgrowth in control (green), blebbistatin (blue) and calyculin A (red)-treated monolayers showing followers per leader reaching a plateau in phase-2. Both Followers per leader and transition time changes upon changing the force correlation length by drugs. **e** Simulating leader cell formation in phase 1 (left) and in phase 2 (right). In phase 1, the relative horizontal distance ($X_C - X_B$) is calculated upon varying $d_{LL}$ where $X_C$ is the horizontal displacement of the boundary between the leader cells and $X_B$ is the baseline displacement. When two leader cells are distinguishable, $X_B \approx X_C$ as shown in the overlay with experimental data. In phase 2, an advanced cellular cohort guided by the leader cell is shown by a circle of radius R attached to the remaining bulk. The effect of a point force $\mathbf{F_L}$ exerted by the leader on the remaining bulk is measured, i.e., the relative displacement of the bulk center $U_C$ with respect to the leader cell displacement $U_L$. **f** Relative horizontal displacement $|X_C - X_B|$ plotted against distance $d_{LL}$ and relative displacement $U_C/U_L$ plotted against distance R. With distance, both $|X_C - X_B|$ and $U_C/U_L$ decay exponentially with a characteristic decay length equivalent to the force localization length $L_P$. Modification of cell stiffness modifies $L_P$ (dotted lines) and therefore the characteristic decay length of the curves changes upon tuning the cell stiffness with drugs. NS: not significant, $P > 0.05$, Mann–Whitney test. Scale bars, 100 μm

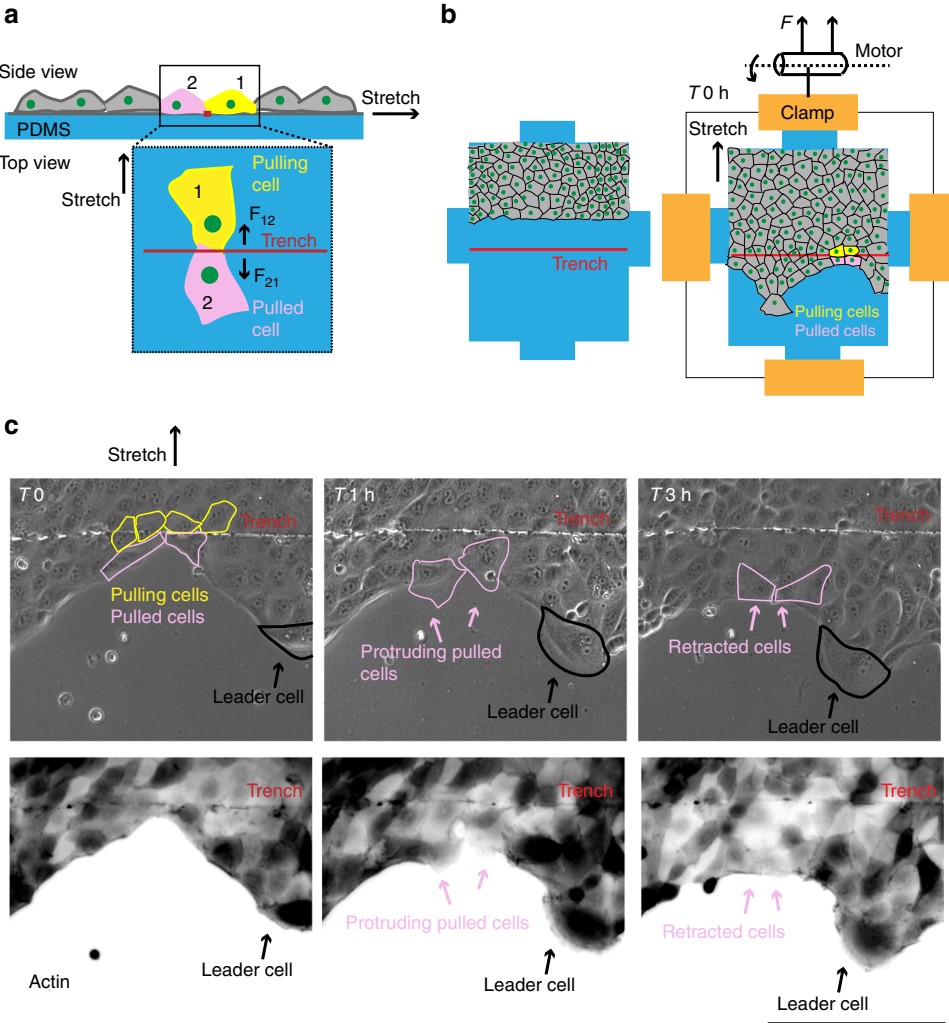

**Fig. 5** Followers pull on the future leader cells. **a** Schematic illustration showing side view and top view of the cell-pulling experiment. Cells seeded on PDMS substrates (in blue) with a fine trench (in red). Upon uniaxial stretching, cells on one side of the trench exert pulling forces on cells on the other side ($F_{12}$ and $F_{21}$). **b** Cell monolayer seeded on the PDMS substrate containing a fine trench (in red) about 100 μm away from the cells, confinement was released and cells were allowed to migrate until one row of cells cross the trench at various locations (right panel). **c** Representative time lapse phase contrast images (top panel) and actin images (bottom panel) showing cells moving on PDMS substrate with a trench. Upon stretching, cells ahead of the trench protrude due to the pulling force from behind (middle panel). These protruding cells however retract, likely due to the existing pulling force coming from the leader at the tip of the outgrowth (right panel). Scale bar = 100 μm

microscope (Olympus CKX53). The PDMS chamber was then coated with 10 µg/ml fibronectin for 1 h at 37°C. Cells were then cultured until confluency on this fibronectin-coated membrane under confined conditions such that the trench is between 100 and 200 µm away from the cells (Fig. 5a). Next day, the confinement was released and cells were allowed to migrate until they have crossed the trench at several locations. The Cell-pulling experiment was then carried out by stretching the membrane unidirectionally with an impulse strain (25% per second). Cells were kept in the pulled condition for 2 min and are then relaxed. The PDMS membrane was immediately taken to the microscope for time lapse imaging.

**Modeling and simulation**. The epithelial layer of height $h_c$ was modeled as a thin elastic layer of elasticity $E_C$, Poisson's ratio $v$ and an isotropic contraction stress $\sigma_0$. The layer is further coupled to the underlying substrate elastically via springs of stiffness density $Y$[31]. The force balance equation results:

$$\sigma_{ij,j} - Yu_i = 0$$

Following constitutive relation, with a linearized strain $\epsilon_{ij} = \frac{1}{2}(u_{ij} + u_{ji})$ was used:

$$\sigma_{ij} = 2\mu\epsilon_{ij} + (\lambda\epsilon_{kk} + \sigma_0)\delta_{ij}$$

with the two-dimensional Lame' coefficients $\lambda = \frac{hE_c v}{1 - v^2}$ and

$$\mu = \frac{hE_c}{2(1 + v)}$$

We assumed a constant active stress $\sigma_0$ throughout the cell layer, which leads to $\sigma_{0,i} = 0$ within the cell layer. The active contraction manifests itself as the remaining stress normal to the exterior boundary of the layer and will be introduced via the boundary conditions. The force balance equation now simplifies to:

$$\sigma_{ij,j} = \lambda u_{k,ki} + \mu\left(u_{i,jj} + u_{j,ij}\right) = Yu_i$$

With the three-dimensional incompressibility condition $v \approx 0.5$, we arrive at: $u_{k,ki} + \frac{1}{2}\left(u_{i,jj} + u_{j,ij}\right) = \frac{u_i}{L_P^2}$, where $L_P$ is the localization length, $L_P = \sqrt{\frac{hE_c}{Y(1+v)}}$ and can be interpreted as the length up to which a point force is transmitted. To distinguish the contributions from bulk and substrate parameters, the localization length is defined, as described previously[32]:

$$L_P = \sqrt{L_a^2 + L_s^2}$$

with localization length due to the action of focal adhesions

$$L_a = \sqrt{\frac{E_c h_c L l_{c0}}{k_a}}$$

and due to the substrate

$$L_s = \sqrt{\frac{E_c h_c L l_{c0}}{\pi E_s\left(\frac{1}{h, 2\pi(1+v_s)} + \frac{1}{L}\right)}}$$

The localization length, $L_P$ was computed by using the default values for the cell layer (Supplementary Table 1), as described previously[34,41]. We used the finite element solver FEniCS to calculate the displacements of the monolayer subject to external stresses[42]. Leader cell formation was simulated by decoupling a part of the monolayer from the substrate, then pulling on it by the action of two Gaussian-shaped forces of width of a finite element mesh size, as approximation for two point forces, separated by the distance $d_{LL}$. Afterwards the full layer was again connected elastically and allowed to contract isotropically. During simulations, Blebbistatin and Calyculin-A treatments were mimicked by modifying bulk parameters as described previously (supplementary table 2)[43,44]. The values of $E_C$ were used as estimates to find a localization length in the range of $d_{LL}$. Relative horizontal displacement ($X_C - X_B$) and relative displacement of $U_C$ ($U_C/U_L$) were plotted as exponential functions of distance. To compute the spatial extent of followers, velocity fields were derived by means of optical flow and Euclidean distance was calculated by comparing the velocity of leader cells from follower cells.

**Statistical analysis**. Statistical analysis was carried out in Prism. All the data underwent normality test in Prism. Statistical significance was calculated by Student's t-test for data following normal distribution and by Mann–Whitney test for the non-normal data. All values were given as mean ± s.d. (standard deviation) or s.

e.m. or shown as boxplots. P-values greater than 0.05 were considered to be statistically not significant.

**Data and code availability**. Relevant data and codes are available from authors upon request.

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

## Acknowledgements

Parts of the research leading to these results have received funding from the European Research Council/ ERC Grant Agreement No. 294852, SynAd. This work is also part of the MaxSynBio consortium, which is jointly funded by the Federal Ministry of Education and Research of Germany and the Max Planck Society. Support was also granted from the Gottfried-Wilhelm-Leibniz Award of the German Science Foundation (DFG). J.P.S. is the Weston Visiting Professor at the Weizmann Institute of Science. J.P.S. and U.S.S. are members of the cluster of excellence CellNetworks at Heidelberg University. We acknowledge support from the Max Planck Society.

## Author contributions

T.D. and J.P.S. conceived the project. M.V., T.D. and J.P.S. designed experiments. M.V. performed all experiments except the elasticity experiments and experiments with HaCaT cells, which were performed by J.D.R. Theoretical model was contributed by D.P. and U.S.S. M.V., D.P., U.S.S., T.D. and J.P.S. analyzed and interpreted experimental data. T.D., D.P., and U.S.S., developed computational tools for traction force and monolayer stress analysis. J.P.S., T.D. and U.S.S. supervised the project. T.D., M.V. and J.P.S. developed and wrote the manuscript with help from D.P. and U.S.S. All authors discussed and commented on the manuscript.

## Additional information

**Competing interests:** The authors declare no competing interests.

