## [Peer Review File · Nature Communications]

Editorial Note: This manuscript has been previously reviewed at another journal that is not operating a transparent peer review scheme. This document only contains reviewer comments and rebuttal letters for versions considered at Nature Communications. Mentions of prior referee reports have been redacted.

REVIEWERS' COMMENTS:

Reviewer #1 (Remarks to the Author):

I am satisfied with the author's response. I recommend publication in Nature Communications.

Reviewer #2 (Remarks to the Author):

This work provides interesting results regarding the dynamics of the cells in the monolayer in the regions near the free edge where the phenomena of "leader-cell" formation occurs. The experiments are interesting, but I think they can greatly improve the interpretation of their results:

1) The authors do not offer a mechanism by which, according to their claim, a leader-cell inhibits somehow the formation of another nearby leader cell. This inhibition, is providing the natural length-scale for the leader-leader separation.

However, as they show in Movie "Video - Biphasic behaviour in a migrating outgrowth", there is in fact a tendency for the tip of the "finger" to split, with a new leader forming very close to the existing one. Such a tip-splitting phenomenon was in fact described both theoretically and experimentally in [17,18], and follows from the curvature-motility feedback.

2) There could be a nice combining of the curvature-motility feedback model, and the discovery by the authors of the importance of the pull of the following cells: In the authors' previous publication [20] they showed that highly curved edge cells are also more strongly and directionally pulled compared to cells at flat edges. One can therefore suggest one possible realization of the proposed curvature-motility feedback, which was originally offered in [17,18] as a phenomenological relation, based on the fact that as cells curve out more they feel a larger pulling force from their back, and this could provide part of the positive feedback needed to induce the spontaneous formation of leader-cells. An initially flat edge will initiate leaders due to fluctuations in the motility and pulling forces behind the edge cells, and fluctuations in the curvature a of the edge cells. The feedback between these two, can possibly drive the initiation and growth of leaders and the "fingers" that they form.

Otherwise the authors lack any feedback mechanism, and without such a mechanism they can not explain the spontaneous initiation of leader-cells. Their suggestion that the length-scale of elastic force decay in the monolayer determines the leader-leader separation remains as purely suggestive, since its not clear at all how one leads to the other.

As I suggest here above, there can in fact be a novel and interesting mechanism proposed based on their observations.

3) The fact that the leader-leader separation is robust to a shape perturbation of the edge is not surprising, and does not preclude a curvature-motility feedback: in the feedback mechanism there is also a spontaneous selection of a typical distance between adjacent leaders, which depends on the intrinsic parameters of the monolayer, and therefore eventually dominates over long times.

4) The inverse relation between speed and velocity-correlation length shown in Fig.S4 reminds me of a similar relation studied in:

Garcia, Simon, et al. "Physics of active jamming during collective cellular motion in a monolayer."

Proceedings of the National Academy of Sciences 112.50 (2015): 15314-15319.

5) The drug Blebbistatin reduces the contractile forces, so should decrease greatly the pull exerted by follower cells on the edge cells. Yet, many more leaders are initiated. How is this compatible with the pulling force being the only route to forming leader cells ? On the other hand, Blebbistatin was observed to cause cells to form more and larger lamellipodia, making them more motile overall (on 2D surfaces). A higher motility of the edge cells can explain the higher density of leaders.

What this result shows is that there are, as we know very well also from single-cell motility, several routes to polarize a cell and make it highly motile. The cells at the edge, that could become leaders, can therefore be polarized by several mechanisms: by being pulled from the back, by being promoted to form larger outwards lamellipodia, etc.

As I noted in point (2), eventually in order to initiate and grow a finger that has a leader at its tip, we need to identify a positive feedback mechanism that ends up providing the edge cell (leader) with higher sustained outwards polarization and motility. How this can be achieved ? the authors identify here the pull at the back of the edge cells as a novel mechanism that plays a role in this feedback. But it not likely to be the only mechanism.

Reviewer #3 (Remarks to the Author):

I am satisfied with the authors' response and support publication. My only request is that the data with the second NF2 siRNA is included in the supplementary figures.

We thank the referees for their helpful comments in the process of revising our manuscript. Below we answer in detail to the referee comments.

Referee 1 (Remarks to the Author):

I am satisfied with the author's response. I recommend publication in Nature Communications.

We thank the reviewer for supporting publication of our findings.

Reviewer #2 (Remarks to the Author):

This work provides interesting results regarding the dynamics of the cells in the monolayer in the regions near the free edge where the phenomena of "leader-cell" formation occurs. The experiments are interesting, but I think they can greatly improve the interpretation of their results:

1) The authors do not offer a mechanism by which, according to their claim, a leader-cell inhibits somehow the formation of another nearby leader cell. This inhibition, is providing the natural length-scale for the leader-leader separation.

However, as they show in Movie "Video - Biphasic behaviour in a migrating outgrowth", there is in fact a tendency for the tip of the "finger" to split, with a new leader forming very close to the existing one. Such a tip-splitting phenomenon was in fact described both theoretically and experimentally in [17,18], and follows from the curvature-motility feedback.

We thank the reviewer for the interesting remark, which is related to Figure 4 of our paper. Using theoretical modelling and simulation, there we give a physical-mechanistic explanation to our experimentally observed results concerning the territory of leader cells. In fact we find a robust distance between the leaders both in phase-1 and in phase-2, when tip splitting occurs. We have reported that an existing leader cell has an ability to transmit force up-to the force localization length (L_P) and any cell that would be contained within this length L_P from a leader cell would be classified as a follower and should therefore only move in the direction of the respective leader cell (Fig. 4). Consequently, in phase-1, the distance between leaders is similar to the length L_P , and in Phase-2, when the outgrowth grows bigger, a new leader is formed when the extent of followers exceeds L_P . Thus the mechanism we report here does explain the observed monolayer dynamics without requiring a dynamic instability of the interface. This does not mean that we exclude the curvature motility feedback hypothesis suggested by the reviewer, but it is not required to explain our observations. In fact the two concepts are not mutually exclusive. However, the mechanical interactions do seem to be upstream of the curvature motility feedback (from Fig. 2e of our paper). If and how the collective is fed by curvature motility remain a question that should be addressed in a future work. We have also included a detailed discussion in the manuscript to address this part (Page 12).

2) There could be a nice combining of the curvature-motility feedback model, and the discovery by the authors of the importance of the pull of the following cells: In the authors' previous publication [20] they showed that highly curved edge cells are also more strongly and directionally pulled compared to cells at flat edges. One can therefore suggest one possible realization of the proposed curvature-motility feedback, which was originally offered in [17,18] as a phenomenological relation, based on the fact that as cells curve out more they feel a larger pulling force from their back, and this could provide part of the positive

feedback needed to induce the spontaneous formation of leader-cells. An initially flat edge will initiate leaders due to fluctuations in the motility and pulling forces behind the edge cells, and fluctuations in the curvature α of the edge cells. The feedback between these two, can possibly drive the initiation and growth of leaders and the "fingers" that they form. Otherwise the authors lack any feedback mechanism, and without such a mechanism they cannot explain the spontaneous initiation of leader-cells. Their suggestion that the length-scale of elastic force decay in the monolayer determines the leader-leader separation remains as purely suggestive, since it's not clear at all how one leads to the other. As I suggest here above, there can in fact be a novel and interesting mechanism proposed based on their observations.

We thank reviewer for this interesting comment and agree that feedback mechanisms will be essential for a complete understanding. We agree that the cells at the highly curved regions should experience more pull and therefore end up becoming leaders. However, this presupposes a mechanical interaction with the bulk, exactly as described here. Also, the fact that not all the leaders at these highly curved regions are necessarily maintained as leaders and the distance between leaders remain similar to an initially unbiased edge (Fig. 2e) shows that cellular interactions and mechanical forces play an integral role in selecting and maintaining the leaders. However, we do not deny the importance of curvature sensing and subsequent feedback in motility and agree with the reviewer that a more comprehensive mechanism that combines the two would be more appropriate. But such a study requires designing new set of experiments and analysis that remain out of the scope of the current study. We now point out these important avenues for future work in the discussion.

3) The fact that the leader-leader separation is robust to a shape perturbation of the edge is not surprising, and does not preclude a curvature-motility feedback: in the feedback mechanism there is also a spontaneous selection of a typical distance between adjacent leaders, which depends on the intrinsic parameters of the monolayer, and therefore eventually dominates over long times.

As mentioned above for comment 1 and 2, we do not preclude the curvature-motility feedback. We understand that the model suggesting curvature motility feedback indicated the possibility that the distance between leaders depend upon the intrinsic properties of the monolayer. However, the nature of the contribution from the monolayer remained mostly unknown, as the events occurring at the onset of or preceding the leader cell formation were not explored in any of the previous works, which is exactly what we provide here.

*4) The inverse relation between speed and velocity-correlation length shown in Fig.S4 reminds me of a similar relation studied in: Garcia, Simon, et al. "Physics of active jamming during collective cellular motion in a monolayer." *Proceedings of the National Academy of Sciences* 112.50 (2015): 15314-15319.*

We thank the reviewer for pointing this out and we now cite this paper in the relevant text.

5) What this result shows is that there are, as we know very well also from single-cell motility, several routes to polarize a cell and make it highly motile. The cells at the edge, that could become leaders, can therefore be polarized by several mechanisms: by being pulled from the back, by being promoted to form larger outwards lamellipodia, etc. As I noted in point (2), eventually in order to initiate and grow a finger that has a leader at its tip, we need to identify a positive feedback mechanism that ends up providing the edge cell (leader) with higher sustained outwards polarization and motility. How this is can be

achieved ? the authors identify here the pull at the back of the edge cells as a novel mechanism that plays a role in this feedback. But it not likely to be the only mechanism.

We thank the reviewer for this comment. We agree that for a leader cell is polarized much more than the non-leader edge cells, and that there can be several routes to polarize a cell to make it a leader. We would like to point out that we do not exclude the possibility of other routes (especially molecular routes by which a specific type of cell polarity is achieved). However, because the epithelial cells are both mechanically and biochemically integrated with their neighbours, we believe that the mechanical and molecular routes greatly influence each other and are not mutually exclusive. What is this molecular route, if it comes upstream or downstream of the mechanical route suggested here and how the two routes influence each other, begs for a new project and therefore remain out of the scope of this study. Again we now point out this challenge in the discussion.

The drug Blebbistatin reduces the contractile forces, so should decrease greatly the pull exerted by follower cells on the edge cells. Yet, many more leaders are initiated. How is this compatible with the pulling force being the only route to forming leader cells? On the other hand, Blebbistatin was observed to cause cells to form more and larger lamellipodia, making them more motile overall (on 2D surfaces). A higher motility of the edge cells can explain the higher density of leaders.

We do not see higher motility of edge cells in blebbistatin-treated monolayer (results not shown). Instead, the fact that force correlation length in blebbistatin-treated monolayer matches with the distance between the leaders (Fig. 2a, 2b) demonstrate that there is a mechanical component to it and that the inter-leader distance in this case is lower due to the fact that leader cell can pull less number of followers.

Furthermore, we have performed a screening of different concentrations of blebbistatin (Supplementary Fig. 8) to choose the concentration of blebbistatin carefully low (5uM) make sure that we are not greatly affecting other signalling pathways.

As I noted in point (2), eventually in order to initiate and grow a finger that has a leader at its tip, we need to identify a positive feedback mechanism that ends up providing the edge cell (leader) with higher sustained outwards polarization and motility. How this is can be achieved ? the authors identify here the pull at the back of the edge cells as a novel mechanism that plays a role in this feedback. But it not likely to be the only mechanism.

We agree with the reviewer and as stated above in point (2) and point (5)- we do not exclude the possibility that feedback between interfacial curvature and motility will further stabilize leader cell identity. Our results, however, support a view in which such effects are downstream of the mechanical selection of leader cells by their future followers (Figure 2e, Figure 5, Supplementary movie 6, Supplementary movie 8, Supplementary movie 10, Supplementary movie 11). Furthermore, upstream or downstream to the biophysical route that we provide here, there might exist molecular routes which are involved in cellular polarization and leader cell formation, that still have to be discovered and is left for future works. We have also included a detailed discussion in this regard in the manuscript.

Reviewer #3 (Remarks to the Author):

I am satisfied with the authors' response and support publication. My only request is that the data with the second NF2 siRNA is included in the supplementary figures.

We thank the reviewer for supporting publication. We have moved the corresponding figure with the second *siRNA* from the response letter to supplementary Fig. 5 (d).